# Novel Inositol 1,4,5-Trisphosphate Receptor Inhibitor Antagonizes Hepatic Stellate Cell Activation: A Potential Drug to Treat Liver Fibrosis

**DOI:** 10.3390/cells13090765

**Published:** 2024-04-30

**Authors:** Natalia Smith-Cortinez, Janette Heegsma, Masa Podunavac, Armen Zakarian, J. César Cardenas, Klaas Nico Faber

**Affiliations:** 1Department of Gastroenterology and Hepatology, University of Groningen, University Medical Center Groningen (UMCG), 9713 GZ Groningen, The Netherlands; 2Department of Chemistry and Biochemistry, University of California, Oakland, CA 94607, USA; 3Center for Integrative Biology, Universidad Mayor, Santiago 7510041, Chile; 4Buck Institute for Research on Aging, Novato, CA 94945, USA

**Keywords:** hepatic stellate cells, calcium, liver fibrosis, mitochondrial metabolism

## Abstract

Liver fibrosis, characterized by excessive extracellular matrix (ECM) deposition, can progress to cirrhosis and increases the risk of liver cancer. Hepatic stellate cells (HSCs) play a pivotal role in fibrosis progression, transitioning from a quiescent to activated state upon liver injury, wherein they proliferate, migrate, and produce ECM. Calcium signaling, involving the inositol 1,4,5-trisphosphate receptor (IP3R), regulates HSC activation. This study investigated the efficacy of a novel IP3R inhibitor, desmethylxestospongin B (dmXeB), in preventing HSC activation. Freshly isolated rat HSCs were activated in vitro in the presence of varying dmXeB concentrations. The dmXeB effectively inhibited HSC proliferation, migration, and expression of fibrosis markers without toxicity to the primary rat hepatocytes or human liver organoids. Furthermore, dmXeB preserved the quiescent phenotype of HSCs marked by retained vitamin A storage. Mechanistically, dmXeB suppressed mitochondrial respiration in activated HSCs while enhancing glycolytic activity. Notably, methyl pyruvate, dimethyl α-ketoglutarate, and nucleoside supplementation all individually restored HSC proliferation despite dmXeB treatment. Overall, dmXeB demonstrates promising anti-fibrotic effects by inhibiting HSC activation via IP3R antagonism without adverse effects on other liver cells. These findings highlight dmXeB as a potential therapeutic agent for liver fibrosis treatment, offering a targeted approach to mitigate liver fibrosis progression and its associated complications.

## 1. Introduction

Liver fibrosis is characterized by the accumulation of excessive scar tissue as a consequence of chronic damage to the liver. This process occurs in most etiologies of chronic liver diseases, such as hepatitis C virus infection, alcohol consumption, and nonalcoholic steatohepatitis [1]. Advanced liver fibrosis results in cirrhosis, which is characterized by disrupted liver architecture due to the accumulation of extracellular matrix (ECM), the presence of proliferative hepatocyte nodules, hepatocellular dysfunction, and increased intrahepatic resistance to blood flow, resulting in hepatic insufficiency and liver function failure [1]. Patients with cirrhosis are likely to develop life-threatening complications, including portal hypertension, hepatic failure, and the development of hepatocellular carcinoma. Liver transplantation is the only effective therapy for end-stage cirrhosis [2]. Contrary to what was believed in the past, liver fibrosis is reversible in its initial stages [2], so establishing anti-fibrotic therapies is key to preventing liver cirrhosis.

The central cell type causing liver fibrosis is the hepatic stellate cell (HSC). These cells are responsible for producing most of the ECM proteins that accumulate in the chronically injured liver. HSCs in the non-diseased quiescent state (qHSC) reside in the space of Disse, store vitamin A in lipid droplets, and characteristically express high levels of the lipid-activated transcription factor peroxisome proliferator-activated receptor gamma (PPAR-γ) [3]. qHSCs regulate retinol metabolism and storage [4] and upon liver injury, release the lipid content, decrease the PPAR-γ expression, and transdifferentiate into myofibroblast-like cells via transforming growth factor beta (TGFβ) signaling [5]. When HSCs transdifferentiate, they start to proliferate, become contractile, and produce excessive ECM proteins like collagen I and III [6], and they are called activated hepatic stellate cells (aHSCs). aHSCs express high levels of alpha smooth muscle actin (α-SMA) and are capable of releasing pro-inflammatory and pro-fibrogenic cytokines like TGFβ [3,7]. Furthermore, the mitochondrial and glycolytic metabolisms strongly increase during HSC activation to support the energetic demands that these highly proliferative, contractile, and migratory cells need [8,9,10,11].

The role of Ca^2+^ in the activation of HSCs has been investigated in detail. The endoplasmic reticulum (ER)-resident Ca^2+^ channel inositol 1,4,5-trisphosphate receptor (IP3R) type I translocates from a perinuclear location in qHSCs to (SMA+) microfilaments in the cell extensions of aHSCs, where it colocalizes with the ER marker calreticulin and regulates cell contraction [12]. Lowering the intracellular Ca^2+^ concentration [12,13,14,15] or inhibiting the transport of Ca^2+^ into the mitochondria [16] reduces the proliferation, activation, migration, and contraction of HSCs. Mechanistically, lowering Ca^2+^ in mitochondria has been shown to reduce mitochondrial metabolism [17], inhibit mitochondrial fusion [18], reduce ATP levels, and affect cell cycle progression [19].

Xestospongin B is a highly selective membrane-permeable, IP3R-competitive inhibitor derived from a marine sponge [20], described to selectively kill cancer cells by lowering mitochondrial metabolism [19]. A new synthetic variant of Xestospongin B, desmethylxestospongin B (dmXeB), has been shown to have the same inhibitory effects on ER-to-mitochondria calcium transfer and to selectively induce cancer cell death similarly to Xestospongin B [21]. Because intracellular/mitochondrial Ca^2+^ is necessary to maintain the aHSC phenotype (proliferation, αSMA expression, contraction), and because dmXeB potently and specifically blocks IP3R-mediated Ca^2+^ signaling, we evaluated the capacity of dmXeB to inhibit HSC activation and thus to be a candidate drug to treat liver fibrosis.

## 2. Materials and Methods

### 2.1. Reagents

Desmethylxestospongin B (dmXeB) (0–10 µmol/L), Ru360 (0–10 µmol/L, Sigma-Aldrich, San Luis, MO, USA), dimethyl α-ketoglutarate (5 mmol/L, Sigma-Aldrich), methyl pyruvate (5 mmol/L, Sigma-Aldrich), nucleosides mix (adenosine, cytosine, guanosine, uridine, and thymidine, 100 µmol/L each, Sigma-Aldrich).

### 2.2. Rat HSC Isolation, Cell Culture, and Treatments

Primary rat HSCs were isolated by a two-step perfusion of the liver with pronase (Merck, Amsterdam, The Netherlands) and collagenase-P (Roche, Almere, The Netherlands). Afterward, cells were separated using Nycodenz (Axis-ShieldPOC, Oslo, Norway) gradient centrifugation, as described before. Freshly isolated qHSCs were culture-activated to aHSCs for 7 days. Cells were treated 4 h or 7 days after isolation. Cells were grown in 5% CO_2_ and 37 °C in ambient air.

Primary rat hepatocytes were isolated from male Wistar rats using a two-step collagenase perfusion to the liver, as described previously [22], and cultured in William’s E medium (Invitrogen, Breda, The Netherlands) at 37 °C in 5% CO_2_. Experiments were initiated 4 h after isolation.

LX-2 cells (Merk Millipore (SCC064), Burlington, MA, USA), and used in passages 22–30. Dulbecco’s Modified Eagle’s Medium (DMEM) with high glucose (Sigma-Aldrich), supplemented with 10% fetal calf serum (Sigma-Aldrich), and 1% antibiotics (Sigma-Aldrich), was used for culturing them. Cells were passed with trypsinization, and medium was refreshed every 3 days or when necessary. Cells were grown in 5% CO_2_ and 37 °C in ambient air.

### 2.3. Human Liver Organoid Culture

Human liver organoids were established as described previously [23] from healthy livers after rejection for transplantation. The liver specimens for this research were obtained according to Dutch legislation and the code of conduct for responsibly dealing with human-derived material in health research. The collection and use of human livers were approved by the Medical Ethical Committee of the University Medical Center Groningen (UMCG) on 25 August 2014 in the license number 2014-77. The use of coded-anonymous human tissue eliminated the need for written consent for “further use” of human material. Liver tissue was cut into small pieces (1–2 mm^3^) and digested with digestion solution containing 2.5 mg/mL of collagenase D (Sigma-Aldrich) containing 0.1 mg/mL of DNAse I (Roche, Almere, The Netherlands) in HBSS (without Ca^2+^ and Mg^2+^) (Thermo Fisher, Waltham, MA, USA). After digestion, single cells were washed with DMEM and resuspended in 60% Matrigel Matrix Basement Membrane (BD Bioscience, San Jose, CA, USA). After the Matrigel/cell suspension was solidified, expansion medium was added. Expansion medium consisted of adDMEM/F12 (Thermo Fisher) supplemented with 1× L-glutamine (Lonza, Basel, Switzerland), HEPES 10 mM (Sigma-Aldrich), penicillin–streptomycin–amphotericin B (Lonza), gentamycin (Thermo Fisher), 50% homemade Wnt-3a conditioned medium (L-Wnt3A cells were a kind gift of Dr. R.G.J. Vries, Hubrecht Institute, Utrecht, The Netherlands), 1× B27 (Invitrogen) and 1× N2 (Thermo Fisher), 1.25 mM N-acetylcysteine (Sigma-Aldrich), 100 ng/mL FGF10 (PeproTech, Cranburry, NJ USA), 10 mM nicotinamide (Sigma-Aldrich), 50 ng/mL HGF (PeproTech), 0.1 µg/mL Rspo-1 (Stemcell, Cologne, Germany) and 50 ng/mL EGF (PeproTech), 100 ng/mL Noggin (R&D, Minneapolis, MN, USA), 10 μM Y27632 (Sigma-Aldrich), 0.5 μM A83-02 (Tocris, Bristol, UK), and 10 µM Forskolin (Sigma-Aldrich). Wnt-3a conditioned medium was prepared as described previously. Once a week, the organoids were removed from the Matrigel using mechanical disruption and passed. Organoids were grown in 5% CO_2_ and 37 °C in ambient air.

### 2.4. Real-Time Monitoring of Cell Proliferation

Proliferation was assessed using the xCELLigence system (RTCA DP; ACEA Biosciences, Inc., San Diego, CA, USA). Rat aHSCs were plated in E-plates to record cell proliferation according to the manufacturer’s instructions. Treatment was started 24 h after attachment and finished after 72 h of treatment. Results were recorded and analyzed with RTCA software v1.1.

### 2.5. Real-Time Imaging Monitoring of Cell Dynamics

Cell migration, morphology, and proliferation were assessed with the live-cell analysis system Incucyte (Sartorius|Biopharma, Göttingen, Germany). Primary rat quiescent or activated HSCs were seeded, and after 4 h or 24 h of attachment, cells were treated and taken to the IncuCyte ZOOM^®^ platform, which was housed inside a cell incubator at 37 °C/7.5% CO_2_, for 72 h. Nine images per well from three technical replicates were taken every 2 h using a 10× objective lens and then analyzed using the IncuCyte™ Basic Software v2018A.

### 2.6. RNA Isolation, cDNA Synthesis, and Real-Time Quantitative PCR

Quantitative real-time reverse transcription polymerase chain reaction (qRT-PCR) was performed as previously described [10]. Briefly, RNA was isolated from cell cultures using TRIzol^®^ reagent according to the supplier’s instruction (Thermo Fisher,). RNA quality and quantity were determined using a Nanodrop 2000c UV-vis spectrophotometer (Thermo Fisher). cDNA was synthesized from 2.5 µg RNA using random nanomers and M-MLV reverse transcriptase (Invitrogen). Taqman primers and probes were designed using Primer Express 3.0.1 and are shown in Appendix A. All target genes were amplified using the Q-PCR core kit master mix (Eurogentec, Seraing, Belgium) on a 7900HT Fast Real-Time PCR system (Thermo Fisher). SDSV2.4.1 (Thermo Fisher) software was used to analyze the data. Expression of genes is presented in 2-delta CT and normalized to 18S.

### 2.7. Protein Isolation, Quantification, and Western Blot Analysis

Protein samples were prepared for Western blot analysis as described previously [10]. Protein concentrations were quantified using the Bio-Rad protein assay (Bio-Rad, Hercules, CA, USA) with bovine serum albumin (BSA) as a standard. Samples totaling 30 µg protein were separated on Mini-PROTEAN^®^ TGX™ precast 4–15% gradient gels (Bio-Rad) and transferred to nitrocellulose membranes using the Trans-Blot Turbo transfer system (Bio-Rad, Hercules, CA, USA). Primary antibodies used were α-SMA (mouse monoclonal, Sigma Aldrich, A5228, 1/1000), Collagen-I (goat polyclonal, Southern Biotech, Homewood, AL, USA, 1310-01, 1/1000) and GAPDH (rabbit polyclonal, Calbiochem, CB1001, 1/10,000), and appropriate horseradish peroxidase (HRP)-conjugated secondary antibodies (1:2000; P0448, DAKO, Glostrup, Denmark) were used for detection. Proteins were detected using the Pierce ECL Western blotting kit (Thermo Fisher). Images were captured using the chemidoc XRS system and Image Lab version 3.0 (Bio-Rad).

### 2.8. Monitoring of Mitochondrial Metabolism Using Seahorse

Oxygen consumption rates (OCRs) were measured using a mitochondrial stress kit in the extracellular flux analyzer Seahorse (Agilent Technologies, Santa Clara, CA, USA) following the manufacturer’s instructions and as described previously [10]. Briefly, respiratory chain inhibitors were added sequentially at the indicated time points. OCR was measured at 37 °C.

### 2.9. Immunofluorescence Microscopy

Cells were seeded in 12-well plates containing glass coverslips after isolation or passaging. Coverslips were washed, fixed (4% paraformaldehyde (Sigma-Aldrich), 10 min), and permeabilized (0.1% triton X-100 (Sigma Aldrich), 10 min) prior to non-specific blocking (2% bovine serum albumin (Sigma Aldrich), 30 min). After blocking, coverslips were incubated with the primary antibodies anti-α-SMA (mouse monoclonal, Sigma Aldrich, A5228, 1/100) and anti-Collagen-I (goat polyclonal, Southern Biotech, 1310-01, 1/200) and with secondary antibodies. Coverslips were then mounted with Vectashield Antifade Mounting Medium with DAPI (Vector Laboratories, Gdynia, Poland). Coverslips were air dried, sealed, stored at 4 °C, and covered away from light until further use. Images were obtained in a Zeiss 410 (Oberkochen, Germany) inverted laser scan microscope, with 16× or 40× magnification objectives using immersion oil, and processed using ImageJ software (1.53a 4 May 202).

### 2.10. Sytox Green Nuclear Staining

HSCs were incubated with Sytox green nucleic acid dye (Invitrogen) for 15 min, as described before, to determine necrosis [24]. Images were obtained in a Zeiss 410 inverted laser scan microscope with a 16× magnification objective and processed using ImageJ software.

### 2.11. Statistical Analysis

All data are presented as mean ± standard error of the mean (SEM). Significance of differences between groups was tested using one-way ANOVA. Calculations were made using the software of GraphPad Prism 8. Results were considered statistically different at a *p* value < 0.05.

## 3. Results

### 3.1. DmXeB Decreases aHSC Proliferation, Migration, and Activation

Exposing fully activated primary rat aHSCs to 5 µM dmXeB strongly impaired cell proliferation, leading to a 50% reduction in cell confluency after 72 h (Figure 1A). This was not observed when aHSCs were exposed to lower doses (1.25 and 2.5 µM) of dmXeB. In line, 5 µM dmXeB decreased mRNA levels of both Col1a1 and Acta2; exposure to 1.25 and 2.5 µM dmXeB did not lower the expression of these fibrosis markers when compared with untreated aHSCs (Figure 1B). Concurrent with gene expression, protein levels of collagen I were strongly reduced in 5 µM dmXeB-exposed aHSCs, as analyzed with Western blotting (Figure 1C) and immunofluorescence microscopy (Figure 1D, top panels). Interestingly, 5 µM XeB also strongly reduced the αSMA protein levels in aHSCs, which became virtually undetectable with immune fluorescence microscopy (Figure 1D, bottom panels). Furthermore, 5 µM dmXeB reduced the aHSCs’ motility at 2 h after exposure and persisted until the end (72 h) of the treatment (Appendix A). HSC proliferation became strongly impaired at 7.5 and 10 µM dmXeB, and the highest dose became toxic to primary HSCs (Appendix AA,B).

We additionally treated the human HSC cell line LX-2 with 0–10 µM dmXeB for 72 h and analyzed its proliferation and activation. After treatment, only at the highest concentration of dmXeB used (10 µM) was a decrease in mRNA levels of the activation markers *COL1A1* and *ACTA2* observed (Figure 1E). Collagen-I protein levels showed a tendency to decrease in the 10 µM dmXeB-treated group; however, no statistical significance was detected at this time (Figure 1F). Overall, these results suggest that dmXeB antagonizes hepatic stellate cell activation in both rat and human HSCs.

### 3.2. DmXeB Prevents HSC Activation and Preserves HSC Quiescence

To assess whether dmXeB also prevents HSC transdifferentiation, we initiated dmXeB exposure 4 h after seeding freshly isolated primary rat qHSCs. Cell proliferation and confluency were analyzed in an Incucyte real-time cell imager. As expected, untreated control cells became highly proliferative in the first 48 h, starting at 20% confluency at seeding on day 0 to ~100% confluency at day 6 (Figure 2A: left panel shows selected images at days 0, 2, and 6; the graph shows quantification of confluency). Exposure to 5 µM dmXeB fully blocked the culture-induced proliferation of qHSCs (Figure 2A). Importantly, 5 µM dmXeB was not toxic for qHSCs, as they started to proliferate immediately after the medium was refreshed, thereby ceasing dmXeB exposure to the HSCs (Figure 2A). These dmXeB-pretreated HSCs reached almost full confluency after approximately 4 days of additional culture in a dmXeB-free medium. In line with the effect of culture-induced proliferation, 5 µM dmXeB also significantly suppressed the culture-induced gene expression of Col1a1 and Acta2 compared with the control HSCs (Figure 2B). As expected, culture-activated primary rat HSCs rapidly lost their vitamin A content, which remained more clearly detectable after 48 h of dmXeB treatment (Figure 3C).

### 3.3. Reduction in HSC Activation by dmXeB Is Mediated by Reduced Mitochondrial Respiration

Activated HSCs have a high combined mitochondrial and glycolytic metabolism to support energy-demanding processes like proliferation, migration, and ECM production [10]. Calcium is an essential cofactor for the dehydrogenases’ activity in the TCA cycle, which produce metabolites to support ETC, mitochondrial respiration, and overall cell bioenergetics [25]. We measured the oxygen consumption rate (OCR) to assess mitochondrial metabolism after 2 h of 5 µM dmXeB treatment in activated HSCs. DmXeB strongly reduced the maximal mitochondrial respiration, and a tendency of decreased basal respiration was observed (Figure 3A). To confirm that mitochondrial Ca^2+^ is important to support relevant cellular processes like cell proliferation, HSCs were treated with an inhibitor, e.g., Ru360, of the mitochondrial Ca^2+^ uniporter (MCU), the main transporter responsible for mitochondrial Ca^2+^ uptake. Similar to dmXeB, 5 µM Ru360 suppressed HSC proliferation (Figure 3B). Higher concentrations of Ru360 were toxic to HSCs (Figure 3B). To confirm that dmXeB prevented proliferation by reducing the activity of the TCA cycle dehydrogenase enzymes, we supplemented the medium of dmXeB-treated cells with either methyl-pyruvate or dimethyl-α-ketoglutarate to boost the activity of the enzymes and overcome the inhibition imposed by dmXeB [26]. Methyl-pyruvate and dimethyl-α-ketoglutarate partially rescued the proliferation of the dmXeB-treated HSCs (Figure 3C). As highly proliferative cells, HSCs may rely on the TCA cycle to produce the intermediate metabolites needed to synthesize pyrimidine and purine nucleotides, a phenomenon that partially occurs in mitochondria [27]. To determine whether a shortage of nucleotides is responsible for the effect observed with dmXeB, we also supplemented the medium of dmXeB-treated HSCs with a nucleotide mix (NUC). Indeed, NUC supplementation to dmXeB-treated cells partially enhanced the HSC proliferation (Figure 3C). Overall, these results suggest that dmXeB prevents HSC proliferation, activation, and migration by inhibiting the TCA cycle–Ca^2+^ dependent enzymes, hence reducing mitochondrial respiration and lowering the intermediate metabolites needed for several cellular processes, including cell proliferation.

### 3.4. DmXeB Does Not Affect Hepatocytes’ or Liver Stem Cells’ Viability

Hepatocytes compose approximately 80% of the total cell population in the liver, carry out most of its functions, and do not proliferate in vitro. We treated primary rat hepatocytes to analyze the potential toxicity of dmXeB to other liver cell types, which could limit its therapeutic applicability for liver diseases. First, we analyzed hepatocyte viability with the real-time analyzer xCELLigence. In control conditions, hepatocytes attach to the plate during the first 4 h after seeding, increasing the impedance and hence the cell index. After this initial increase, the impedance drops again and stabilizes over the following 20 h (Figure 4A), which is mostly due to a morphological change in the hepatocytes and not because of cell death. Interestingly, dmXeB did not cause a decrease in impedance after the attachment of the hepatocytes. Moreover, concentrations up to 10 µM dmXeB did not induce hepatocyte cell death (Figure 4A,B). Furthermore, dmXeB (up to 10 µM) did not significantly affect the gene expression of typical hepatocyte markers, such as albumin and the bile salt export pump (BSEP) (Figure 4C).

Additionally, we analyzed the potential toxicity of dmXeB on human liver organoids generated from tissue-resident progenitor cells. Cultured in vitro, these organoids are composed of a mixture of progenitor cells, hepatocytes, and cholangiocytes. Exposure to dmXeB concentrations ranging from 0–5 µM for 72 h did not affect the liver organoid shape (Figure 4D) or proliferation (measured by organoid diameter, Figure 4E), nor the expression of the stem cell marker leucine-rich repeat-containing G-protein coupled receptor 5 (LGR5) (Figure 4F). Although exposure to 10 µM XeB significantly altered liver organoid morphology (Figure 4D) and significantly reduced organoid growth (Figure 4E), it did not affect the gene expression of LGR5 (Figure 4F).

Overall, these results suggest that 5 µM dmXeB is not toxic for hepatocytes and liver progenitor cells.

## 4. Discussion

In this study, we show that the inhibition of IP3R with dmXeB has anti-fibrotic properties in vitro by preventing and reversing HSC activation. Dysregulated IP3R-mediated Ca^2+^ signals have been shown to play an important role in fibrosis by controlling HSC activation, migration, proliferation, and fibrogenesis [12,13,14,15,16]. Here, we demonstrate that inhibiting the IP3R with the novel dmXeB lowers proliferation, migration, and activation markers in primary rat aHSCs. The dmXeB delayed qHSC to aHSC transdifferentiation, maintaining a qHSC phenotype in culture. Interestingly, treated qHSCs retained activation potential, because after removal of the treatment, the HSCs activated normally. Rat primary HSCs are strongly dependent on mitochondrial metabolism to activate and maintain their activation status [8,9,10]. IP3R-mediated Ca^2+^ transfer to the mitochondria is essential for cellular bioenergetics and mitochondrial metabolism [17], given that rate limiting enzymes in the TCA cycle such as pyruvate dehydrogenase, alpha ketoglutarate dehydrogenase, and isocitrate dehydrogenase are Ca^2+^-sensitive [25,26]. Accordingly, it was confirmed that the suppressed mitochondrial metabolism and decreased proliferation in dmXeB-treated aHSCs could be partially reversed by the supplementation of TCA cycle intermediates [25,26]. IP3R-mediated Ca^2+^ transfer to the mitochondria not only controls cellular homeostasis by producing energy and providing cofactors for important enzymes, it also allows for the generation of metabolites needed for the biosynthesis of macromolecules, including pyrimidine and purine nucleotides [27]. Hence, the decrease in aHSC proliferation observed after the addition of dmXeB is partially due to a reduced nucleotide production. Overall, this suggests that dmXeB prevents HSC proliferation and activation by inhibiting Ca^2+^-dependent TCA cycle enzymes and preventing nucleotide synthesis to sustain cell growth, activation, and migration.

All three IP3R isoforms are expressed in the liver. HSCs express only type I IP3R, while hepatocytes express types I and II IP3R, and IP3R type III is found mainly in cholangiocytes [12,28]. In hepatocytes, IP3Rs control glucose secretion and mitochondrial metabolism. Interestingly, in obese mice, types I and II IP3R are upregulated, leading to increased cytoplasmic and mitochondrial Ca^2+^ concentrations and mitochondrial dysfunction [29]. The inhibition of type I IP3R by short-hairpin RNA led to leaner mice and improved mitochondrial function [29]. Surprisingly, an increased expression of type I IP3R was observed in NASH patients, a disease associated with metabolic syndrome, obesity, and liver fibrosis due to mitochondrial dysfunction [30]. In this context, drugs inhibiting IP3R, like dmXeB, have a promising future because they could potentially prevent or treat fatty liver and NASH, in addition to liver fibrosis, by reducing HSC activation.

The role of Ca^2+^ in the transdifferentiation of HSCs has been thoroughly investigated. Specifically, lowering Ca^2+^ entry to mitochondria reduced HSC proliferation, migration, and contraction by reducing the mitochondrial metabolism, inhibiting mitochondrial fusion, reducing ATP levels, and affecting cell cycle progression [16]. These studies targeted mitochondrial Ca^2+^ uptake by inhibiting the MCU in primary rat HSCs and human LX-2 cells. Similarly, by targeting the Ca^2+^ transfer from the ER to mitochondria, dmXeB-treated HSCs reduced proliferation, migration, and activation markers. Ca^2+^ in mitochondria plays a pivotal role in activating various proteins, including pyruvate dehydrogenase and α-ketoglutarate dehydrogenase. Consequently, when Ca^2+^ levels are low, the activity of these enzymes is diminished, resulting in suppression of the entire TCA cycle. To counteract this effect, the addition of 5 mM methyl-pyruvate (pyr) and 5 mM dimethyl-α-ketoglutarate, the substrates of the aforementioned enzymes, can reactivate the enzymes through a mass effect. This normalizes respiration, effectively preventing cell death [17,19,26]. We hypothesize that the mechanism by which dmXeB prevents aHSC proliferation is similar to the one described by Ai et al. [16] because we observed a reduction in mitochondrial respiration after dmXeB treatment and, by supplementing the medium with the mitochondrial intermediates methyl pyruvate and dimethyl α-ketoglutarate, which fuel the TCA cycle, the proliferation of HSCs was restored. IP3R enhances fibroblasts’ activity in other tissues also, increasing collagen production and promoting fibroblast migration [31]. Given these similarities, dmXeB may be an interesting compound not only to treat liver fibrosis, but also fibrosis in other organs.

There are no approved drugs for the treatment of NASH or liver fibrosis, so there is an unmet need to find novel therapeutic interventions for treating these patients. Because of the effect of obesity, metabolic syndrome, and diabetes type 2 on the progression of NAFLD to NASH, several drugs target insulin resistance and steatosis simultaneously. Recently, drugs that target Ca^2+^ signaling by modulating AMPK, the SERCA protein, cAMP, PKC, and SOC channels have been identified as potential targets for the treatment of liver steatosis and insulin resistance [32]. Known antidiabetic drugs (e.g., metformin or rosiglitazone) affect intracellular Ca^2+^ signaling pathways [32]. We have previously shown that metformin reduces activation, proliferation, and migration in human primary HSCs [10], and others have shown that metformin reduces the mortality rate by 57% in diabetic patients with cirrhotic liver disease [33]. Moreover, metformin reverses already established lung fibrosis in mice [34]. Interestingly, other drugs with Ca^2+^ modulation properties, like melatonin and curcumin, have been shown by us and others to reduce HSC activation in vitro and in vivo as well [35], although these compounds also have Ca^2+^-unrelated effects. The effects of Ca^2+^ modulation on inflammatory diseases have been also investigated. For example, it has been described that in hepatitis B infection, the hepatitis B virus can elevate cytosolic Ca^2+^ levels, and the activation of intracellular Ca^2+^ signaling contributes to viral replication via multiple molecular mechanisms. Furthermore, it has been suggested that targeting Ca^2+^ signaling with pharmaceuticals is a potent approach for the treatment of HBV infection [36]. On the other hand, immune cell infiltration is pivotal in fibrosis development, as these cells undergo cytoskeletal alterations to facilitate infiltration. Ca^2+^ is known to regulate cytoskeletal dynamics and migration in various cell types. Consequently, we anticipate that dmXeB may diminish immune cell infiltration, thus potentially contributing to a reduction in liver fibrosis and potentially also aiding viral hepatitis infections.

Xestospongin B is a specific and selective IP3R inhibitor that prevents Ca^2+^ release from the ER mediated by the IP3R [20]. Extracted from a marine sponge, its availability is scarce. A novel chemical synthesis has allowed the generation of dmXeB [21]. DmXeB has a chemical structure similar to Xestospongin B and shows similar effects in reducing IP3R-mediated Ca^2+^ signals and selectivity, and potently kills cancer cells [21,26]. Because of its selectivity in targeting highly proliferating cells, dmXeB holds promise as a therapeutic drug. In the healthy liver, there are limited proliferating cells, if any. Indeed, when dmXeB was added to non-proliferative primary hepatocytes in culture, no toxic effects were observed, thus confirming a specific role in targeting cell proliferation. In the fibrotic liver, the main proliferative cells are aHSCs, and thus, they would be the main targets of dmXeB. We observed cytotoxic effects of the compound at high concentrations in proliferative cells like the HSCs and liver progenitor cells. However, non-proliferative cells such as hepatocytes were not harmed by this concentration. During liver regeneration and acute damage, e.g., after partial hepatectomy (removal of up to 70% of the liver for patients with hepatic neoplasms, most frequently colorectal cancer metastases and primary hepatocellular carcinomas), progenitor cells and hepatocytes become proliferative to recover the loss of mass [37]. To test the effect of dmXeB on other proliferative cell types, we used a human liver organoid model that allowed us to expand human liver progenitor cells in vitro. Liver organoid 3D cultures have been developed in the last 10 years; they maintain key characteristics of the progenitor cells from the liver and are proliferative in culture [23,38,39]. After dmXeB treatment (5 µM), human liver organoids continued to grow and proliferate similarly to the control organoids and maintained the expression of the stem cell marker LGR5, suggesting that dmXeB is neither toxic nor does it lower progenitor cell capacity. However, we observed that high concentrations of dmXeB (10 µM) significantly reduced organoid size, which suggests that Ca^2+^ signaling is contributing to liver progenitor cell proliferation. Interestingly, the expression of LGR5 was not reduced at this concentration. In line, it has been described that IP3R-dependent Ca^2+^ signaling is a pathway necessary for liver regeneration [40]. Hence, targeting the liver with the IP3R inhibitor dmXeB should be firmly established in in vivo models of chronic liver disease first.

In conclusion, dmXeB is a novel, selective, and specific IP3R inhibitor that reduces HSC activation, proliferation, and migration by inhibiting TCA cycle enzymes and reducing mitochondrial respiration without direct toxic effects on other liver cell types. DmXeB is therefore a promising candidate drug to treat liver fibrosis.

## Figures and Tables

**Figure 1 cells-13-00765-f001:**
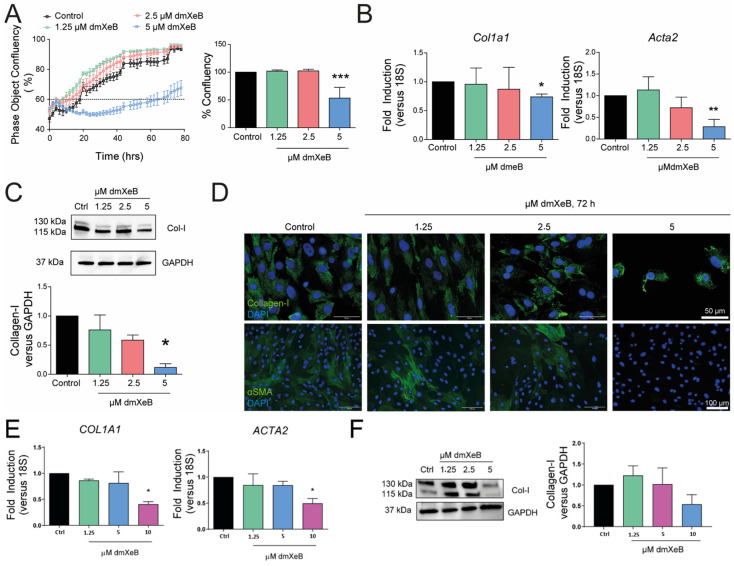
DmXeB prevents HSC proliferation, activation, and migration: Primary rat HSCs were culture-activated in vitro for 7 days and then treated for 72 h with 0–5 µM dmXeB (**A**–**D**); LX-2 cells were treated with 0–10 µM dmXeB for 72 h with 0–10 µM dmXeB (**E**,**F**). (**A**) Proliferation was assessed using real-time automatic imaging (right panel) and confluency was calculated after 72 h of treatment. (**B**) mRNA expression of *Col1a1* (left panel) and *Acta2* (right panel). (**C**) Representative Western blots of collagen type I and GAPDH. Bar graph: Collagen-I/GAPDH and αSMA/GAPDH expressed as average fold change over basal levels. (**D**) Representative images of immunofluorescence microscopy of collagen type I (in green) and DAPI (in blue) (top panels) and αSMA (in green) and DAPI (in blue) (lower panels) of cells treated with dmXeB. Scale bar, 50 µm (top panels) and 100 µm (lower panels) (n = 3). (**E**) mRNA expression of *COL1A1* (left panel) and *ACTA2* (right panel). (**F**) Representative Western blots of collagen type I and GAPDH. Bar graph: collagen/GAPDH expressed as average fold change over basal levels. Scale bar, 50 µm (n = 3). n = 3–4, mean ± SEM, * *p* < 0.05, ** *p* < 0.001, *** *p* < 0.0001 (one-way ANOVA).

**Figure 2 cells-13-00765-f002:**
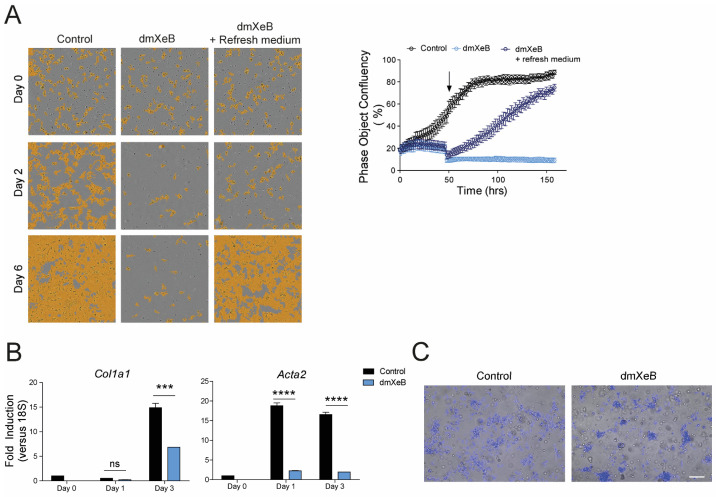
The addition of 5 µM dmXeB maintains HSC quiescency: Freshly isolated rat HSCs were treated with vehicle (control), with 5 µM dmXeB 4 h after isolation for 6 days or with 5 µM dmXeB 4 h after isolation for 2 days followed by a change to a dmXeB-free medium. (**A**) Representative images of day 0-, day 2-, and day 6-treated cells in all conditions assessed with real-time automatic imaging (left panel) and phase object confluency over time (right panel). Black arrow indicates the point when medium was refreshed to dmXeB-free medium. (**B**) mRNA expression of Col1a1 (left panel) and Acta2 (right panel) in control and dmXeB-treated cells for day 0 (4 h after isolation), day 1, and day 3 after treatment. (**C**) Representative images of control and dmXeB-treated cells for 48 h exposed to 405 nm laser in an epifluorescence microscope to measure vitamin A autofluorescence. Scale bar, 50 µm (n = 3). n = 3–4, mean ± SEM, *** *p* < 0.0001, **** *p* < 0.0001 (one-way ANOVA).

**Figure 3 cells-13-00765-f003:**
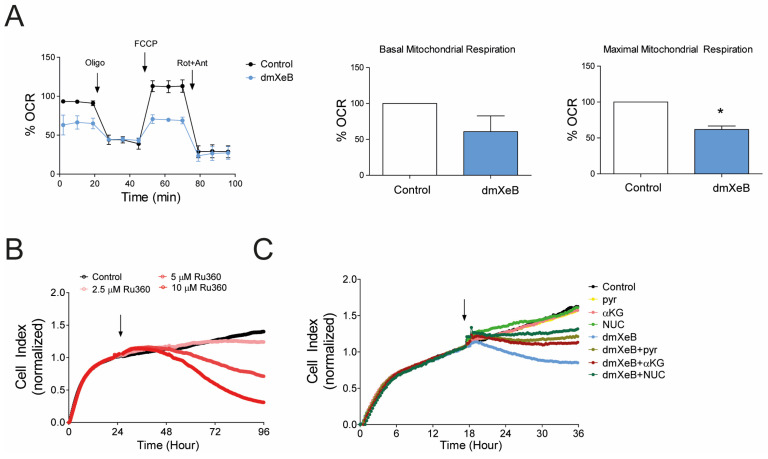
DmXeB reduces mitochondrial respiration by targeting mitochondrial calcium: Primary rat activated HSCs were treated with 5 µM dmXeB as the first injection of a Seahorse experiment 2 h before a mitochondrial stress kit was conducted. (**A**) Oxygen consumption rate profile in basal-, oligomycin-, FCCP-, and rotenone plus antimycin-injected conditions in the control and dmXeB-injected cells (left panel). Basal (middle panel) and maximal (right panel) mitochondrial respiration was calculated. (**B**) The proliferation rate of primary aHSCs treated with 0–10 µM Ru360. (**C**) Representative graph of proliferation rate of primary aHSCs treated with 5 µM dmXeB in combination with 5 mM methyl-pyruvate (pyr), 5 mM dimethyl-α-ketoglutarate (αKG), and 100 µM nucleosides mix (NUC). Arrow represents the start of treatment. n = 3, mean ± SEM, * *p* < 0.05 (one-way ANOVA).

**Figure 4 cells-13-00765-f004:**
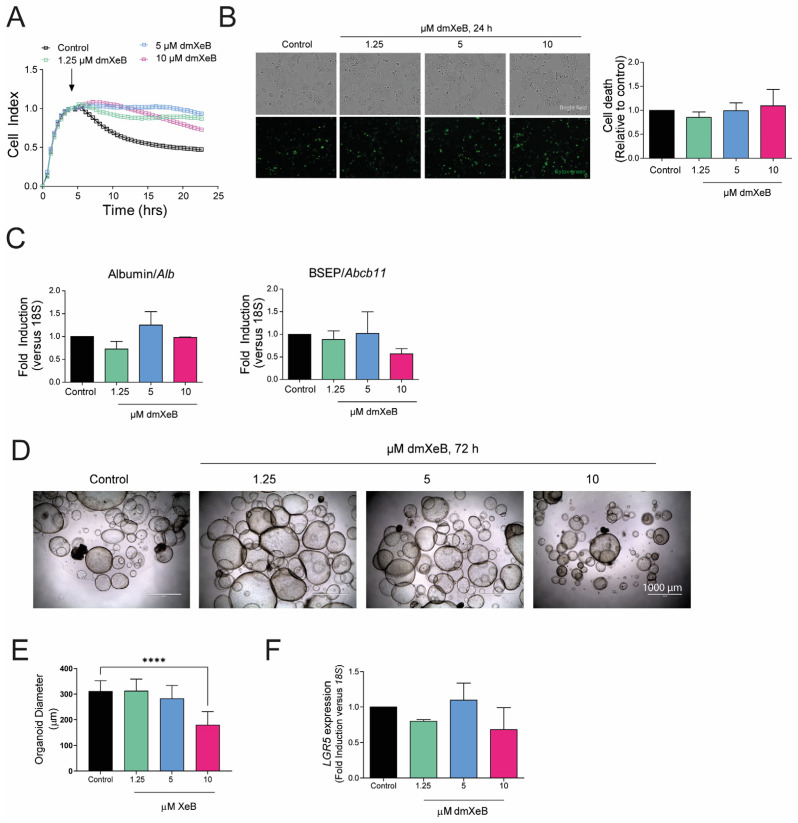
Toxicity of dmXeB to other liver cells: Primary rat hepatocytes were treated with 0–10 µM dmXeB for 24 h (**A**–**C**). (**A**) Cell proliferation rate during 24 h of treatment. Arrow represents the start of treatment. (**B**) Bright field images (upper panel) and Sytox green staining (bottom panel) of hepatocytes treated for 24 h. Bar graph: cell death calculated by quantifying necrotic (stained) nuclei as expressed as fold induction over control (n = 3, mean ± SEM, bar = 100 µm). (**C**) mRNA of albumin and BSEP after 24 h of treatment. Primary human liver organoids were treated with 0–10 µM dmXeB for 24 h (**D**–**F**). (**D**) Bright field pictures of human liver organoids treated for 72 h. (**E**) Diameter of organoids after treatment. (**F**) mRNA expression of LGR5 after treatment. n = 3, mean ± SEM, **** *p* < 0.0001 (two-way ANOVA with Tukey’s multiple comparisons).

## Data Availability

The data generated during the current study are available from the corresponding author on reasonable request.

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
