# Peer review of "Novel Inositol 1,4,5-Trisphosphate Receptor Inhibitor Antagonizes Hepatic Stellate Cell Activation: A Potential Drug to Treat Liver Fibrosis"

_cells, 2024, doi:10.3390/cells13090765_

Round 1

Reviewer 1 Report

Comments and Suggestions for Authors

The experimental study by Smith-Cortinez et al. (cells-2906868) has been designed in order to investigate the efficacy of desmethylxestospongin B (dmXeB), a novel derivative of Xestospongin B, able to selectively inhibits   the inositol 1,4,5-15 trisphosphate receptor (IP3R), in preventing the activation of hepatic stellate cells (HSC). Authors tested dmXeB on freshly isolated rat HSC and provide results indicating that the novel drug inhibits major phenotypical responses operated by HSC, including proliferation, migration and fibrogenesis without affecting HSC vitamin A storage as well as the viability of rat hepatocytes or human liver organoid. Moreover, the compound was shown to suppress mitochondrial respiration in activated HSC but to enhance glycolysis. HSC proliferation inhibited by dmXeB was restored by 22 methyl pyruvate, dimethyl α-ketoglutarate and nucleoside-supplementation. Authors conclude that dmXeB may represent a promising drug that by selectively acting on activated HSC may reduce fibrogenic progression.

The in vitro study by Smith-Cortinez et al. (cells-2906868) is of potential interest and I can offer the following comments.

Major comment

The concept that dmXeB may directly interfere with the process of HSC activation is of potential interest due to the actual lack of validated antifibrotic agents. Although data on organoids are suggestive data presented in the manuscript are essentially “in vitro” (i.e., performed on primary culture of HSC or hepatocytes and, as mentioned, on liver organoids) and it is then reasonable to suppose that Authors will check the real “in vivo” efficacy of the novel derivative of Xestospongin B. If Authors may provide some preliminary data on the effect of dmXeB on an experimental “in vivo”  modelof chronic liver injury this will enhance significantly the message of the study. In addition, Authors should provide additional “in vitro” data to corrobate their overall message.

Specific comments

1. By analysing data provided this reviewer could not find any trace of Suppl. Video 1 which, as for text in the results section, should provide evidence for an effect of dmXbE on HSC migration. Authors should provide the video.

2. The use of human LX2 cells is mentioned in Materials and Methods but this reviewer could not find any data, either described in the text or shown in Figures, related to experiments performed on this immortalized cell line.

3. To further confirm the efficacy of dmXeB on phenotypical responses of activated HSC reported in Figure 1, Authors should use human LX2 cells (as for comment 2 mentioned in Materials and Methods but not apparently used in the study) exposed to PDGF-BB (to test efficacy on GF-induced proliferation) and TGFβ1 (to test efficacy on GF-induced increased synthesis of EXC components).

4. Related to the previous comment: Authors should extend their investigation on more parameters related to fibrosis by analysing at least transcript levels of TGFβ1, PDGF-B, PDGFβR in both primary rat liver HSC and human LX2 cells. It should be of interest also to check whether dmXeB may also affect expression of CCL2, a critical chemokine in chronic liver disease progression which is known to be up-regulated and released also by activated HSC.

5. From Suppl. Figure 1A and 1B one can realize that dmXeB 5 μM is already toxic for HSC and the toxicity seems to start quite rapidly (Suppl. Figure 1A). In the text of Results there is a sentence stating “Concentrations above 5 μM dmXeB 203 became toxic to primary HSC (Supplementary Figure 1A & B)” (see page 5, lines 203-204). This should be clarified and if dmXeB 5 μM is already cytotoxic for aHSC soon after the addition of the drug this should be properly  discussed by Authors. In particular, which is the Author’s view on the dmXeB – related inibition of mitochondrial respiration and induction of cytotoxicity with the          5 μM concentration ?

6.  Are treatment with 5 mM methyl-275 pyruvate (pyr), 5 mM dymethyl-α-ketoglutarate (αKG), and 100 μM nucleosides mix (NUC) able also to prevent (partially) dmXeB cytotoxicity?  

Author Response

Reviwer1

The experimental study by Smith-Cortinez et al. (cells-2906868) has been designed in order to investigate the efficacy of desmethylxestospongin B (dmXeB), a novel derivative of Xestospongin B, able to selectively inhibits   the inositol 1,4,5-15 trisphosphate receptor (IP3R), in preventing the activation of hepatic stellate cells (HSC). Authors tested dmXeB on freshly isolated rat HSC and provide results indicating that the novel drug inhibits major phenotypical responses operated by HSC, including proliferation, migration and fibrogenesis without affecting HSC vitamin A storage as well as the viability of rat hepatocytes or human liver organoid. Moreover, the compound was shown to suppress mitochondrial respiration in activated HSC but to enhance glycolysis. HSC proliferation inhibited by dmXeB was restored by 22 methyl pyruvate, dimethyl α-ketoglutarate and nucleoside-supplementation. Authors conclude that dmXeB may represent a promising drug that by selectively acting on activated HSC may reduce fibrogenic progression.

The in vitro study by Smith-Cortinez et al. (cells-2906868) is of potential interest and I can offer the following comments.

Major comment

The concept that dmXeB may directly interfere with the process of HSC activation is of potential interest due to the actual lack of validated antifibrotic agents. Although data on organoids are suggestive data presented in the manuscript are essentially “in vitro” (i.e., performed on primary culture of HSC or hepatocytes and, as mentioned, on liver organoids) and it is then reasonable to suppose that Authors will check the real “in vivo” efficacy of the novel derivative of Xestospongin B. If Authors may provide some preliminary data on the effect of dmXeB on an experimental “in vivo”  modelof chronic liver injury this will enhance significantly the message of the study. In addition, Authors should provide additional “in vitro” data to corrobate their overall message.

Specific comments

  1. By analysing data provided this reviewer could not find any trace of Suppl. Video 1 which, as for text in the results section, should provide evidence for an effect of dmXbE on HSC migration. Authors should provide the video.

Reply: We apologize for this error. The video has now been added as Supplementary Video 1 online.

  1. The use of human LX2 cells is mentioned in Materials and Methods but this reviewer could not find any data, either described in the text or shown in Figures, related to experiments performed on this immortalized cell line.

Reply: We apologize for this error. The data for the human LX-2 cells was initially excluded. We do appreciate that adding these data is valued by this reviewer and now is included as panels E and F of Figure 1 in the revised manuscript and in lines 216-222 in the text.

  1. To further confirm the efficacy of dmXeB on phenotypical responses of activated HSC reported in Figure 1, Authors should use human LX2 cells (as for comment 2 mentioned in Materials and Methods but not apparently used in the study) exposed to PDGF-BB (to test efficacy on GF-induced proliferation) and TGFβ1 (to test efficacy on GF-induced increased synthesis of EXC components).

Reply: We understand that PDGF-BB and TGFb are relevant factors to study in relation the effect of dmXeB. However, the growth conditions for HSC and LX-2 cells used in this study, e.g. with 20% FCS and cultured on plastic culture plates, already leads to strong (if not maximal) induction of cell proliferation and expression of markers of fibrosis in both primary HSC (Fig. 1A-D) and LX-2 (Fig 1 E-F) cells. To study the effect of PDGF-BB and TGFb, HSC/LX-2 cells first need to be cultured in FCS-free conditions (suppressing proliferation and fibrosis marker gene expression), after which this is stimulated again with either PDGF-BB or TGFb. In this case, we feel that such experiments are redundant to what we already show: dmXeB-mediated suppression of proliferation and activation of HSC and LX-2 cells. Moreover, the limited time for resubmitting our manuscript did not allow us to perform such experiments in time.

  1. Related to the previous comment: Authors should extend their investigation on more parameters related to fibrosis by analysing at least transcript levels of TGFβ1, PDGF-B, PDGFβR in both primary rat liver HSC and human LX2 cells. It should be of interest also to check whether dmXeB may also affect expression of CCL2, a critical chemokine in chronic liver disease progression which is known to be up-regulated and released also by activated HSC.

Reply: We do understand that it would be interesting to include additional analyses of potential signaling factors coming from the HSC and LX-2 cells. However, for this study we were primarily interested in the effector mechanisms for the development of fibrosis, e.g. collagen 1 production and aSMA/ACTA2 expression. These are the gold standard markers for the role of HSC in liver fibrosis. Hence these markers have been used throughout our manuscript. We feel that adding additional markers will not significantly add to the main message of our manuscript, e.g. dmXeB suppresses HSC proliferation and fibrotic potential. And again, the limited time for resubmitting our manuscript did not allow us to perform such experiments in time.

  1. From Suppl. Figure 1A and 1B one can realize that dmXeB 5 μM is already toxic for HSC and the toxicity seems to start quite rapidly (Suppl. Figure 1A). In the text of Results there is a sentence stating “Concentrations above 5 μM dmXeB 203 became toxic to primary HSC (Supplementary Figure 1A & B)” (see page 5, lines 203-204). This should be clarified and if dmXeB 5 μM is already cytotoxic for aHSC soon after the addition of the drug this should be properly  discussed by Authors. In particular, which is the Author’s view on the dmXeB – related inibition of mitochondrial respiration and induction of cytotoxicity with the 5 μM concentration ?

Reply: The inhibition of IP3R-mediated calcium transfer to the mitochondria induced by dmXeB leads to a reduction in mitochondrial respiration, resulting in a state of bioenergetic stress. This stress triggers the activation of autophagy and AMPK, which collectively contribute to maintaining cellular homeostasis in normal cells (PMID: 20655468). Despite the activation of these mechanisms, we consistently observed a modest impact on the viability of normal non-tumorogenic breast cell lines (MCF10), primary fibroblasts, and primary lymphocytes after 24 hours of 5 uM exposure (PMID: 26947070, PMID: 33440859). We hypothesize that this occurs because a small population of cells exhibits higher energetic demands that cannot be met even with the activation of AMPK and autophagy. Additionally, it is possible that the population of cells that died are unable to metabolically switch to glycolysis, leading to their demise.

We have updated the text accordingly in lines 213 to 215 and in lines 413 to 415 of the revised manuscript. Indeed, 5 uM XeB reduced cell proliferation and the cell toxicity assay performed (sytox green staining) shows some necrosis. Still, most cells are alive and stay alive over time as observed in the supplementary videos where they have reduced migration but are still alive. By monitoring cell proliferation, we observed a strong reduction in impedance, suggesting cells are dying at concentrations 7.5 and 10 uM. This was confirmed by sytox green staining where at 10 uM XeB all cells were dead.  Although 5 uM reduces mitochondrial respiration, these cells increase their glycolytic activity to maintain viability, still, because of the reduced metabolites, they are likely less able to proliferate and migrate and produce collagens to the same extent as their non-treated controls.

  1. Are treatment with 5 mM methyl-275 pyruvate (pyr), 5 mM dymethyl-α-ketoglutarate (αKG), and 100 μM nucleosides mix (NUC) able also to prevent (partially) dmXeB cytotoxicity?  

Reply: Yes, they can, as we have shown in previous studies (PMID: 20655468, PMID: 26947070, PMID: 32665411). Ca2+ in the mitochondria play a pivotal role in activating various proteins, including pyruvate dehydrogenase and α-ketoglutarate dehydrogenase. Consequently, when Ca2+ levels are low, the activity of these enzymes is diminished, resulting in a slowdown of the entire TCA cycle. To counteract this effect, the addition of 5 mM methyl-pyruvate (pyr) and 5 mM dimethyl-α-ketoglutarate, the substrates of the aforementioned enzymes, can reactivate the enzymes through a mass effect. This normalize respiration effectively preventing cell death. We have discussed about this role in line 373 of the revised manuscript.

Mitochondrial function is paramount for nucleotide synthesis. When Ca2+ are unavailable and mitochondrial function is compromised, proliferative cells either exit the cell cycle as normal cells do, or undergo mitotic catastrophe like cancer cells, a process that can be prevented by supplying nucleotides. We have not specifically analyzed the effect of pyr, a-KG or NUC supplementation of HSC toxicity, but we did observe they rescue cell proliferation by evaluating impedance in an xCELLigence real time analyzer (Figure 3C).

Reviewer 2 Report

Comments and Suggestions for Authors

The current study covered all the aspects to examine the effects dmXeB may have against the fibrotic effects of HSC by inhibiting their activation via IP3R antagonism. They showed that dmXeB has an anti-fibrotic impact on HSC cells but has no adverse effects on hepatocytes.

Concerns:

  • While 2.5 uM of dmXeB has no effect, 10 uM results in cell death. How these small windows of the dosages could be applied to in vivo studies and clinical trials is still being determined.
  • Fibrosis triggers inflammation; authors tested dmXeB in healthy conditions but need to test the impact on hepatitis.

-What is the effect of dmXeB on immune cell infiltrations under fibrotic situations? Please discuss this notion.

Author Response

Reviwer2

The current study covered all the aspects to examine the effects dmXeB may have against the fibrotic effects of HSC by inhibiting their activation via IP3R antagonism. They showed that dmXeB has an anti-fibrotic impact on HSC cells but has no adverse effects on hepatocytes.

Concerns:

  1. While 2.5 uM of dmXeB has no effect, 10 uM results in cell death. How these small windows of the dosages could be applied to in vivo studies and clinical trials is still being determined.

Response: We are aware of the small window of work observed in cells cultured in vitro. And we do agree that toxicity and therapeutic value remains to be established in follow-up studies. However, we do anticipate on more favorable outcomes in vivo, at least for potential toxic effects on the liver. Indeed, we conducted a pilot experiment for another manuscript, wherein we intravenously administered doses of 5, 10, or 30 mg/kg dmXeB to mice bearing orthotopic breast cancer. These injections were repeated weekly for a month. The concentrations used in this pilot experiment are notably higher than those employed in cell culture. Nonetheless, we observed no toxicity-related side effects in the mice, and no cytotoxic effects were evident in the liver (see below representative images of three different mice treated with the highest 30 mg/kg) concentration).

We confirmed IP3R inhibition at these concentrations, as evidenced by reduced tumor sizes compared to controls and activation of AMPK. Figure has been included in the word file attached.

  1. Fibrosis triggers inflammation; authors tested dmXeB in healthy conditions but need to test the impact on hepatitis.

Response: Unfortunately, we had not explored the effect of dmXeB in the context of inflammation.  According to the literature Ca2+ plays an important role in hepatitis B virus infection (PMID: 34362380). In particular had been shown that the inhibition of IP3R with a less specific inhibitor such as 2APB, have  a significant inhibitory effect on the virus replication. Thus, we expect that dmXeB will behave similarly, reducing virus replication and improving liver condition.  Similarly, the mitochondria associated membranes (MAMs), where the Ca2+ communication between the IP3R and mitochondria occurs, is important for the replication of the Hepatitis C virus (doi.org/10.1016/j.jhepr.2022.100647). Remarkably, dmXeB changed the structure of MAMs (/doi.org/10.1101/2024.03.31.58747). Thus,  we expect dmXeB to affect the replication of the HCV and improve hepatitis in this scenario. Still, we feel that the effect of dmXeB on hepatitis is a topic beyond our current study focusing on liver fibrosis.

  1. What is the effect of dmXeB on immune cell infiltrations under fibrotic situations? Please discuss this notion.

Response: This is an intriguing consideration. Immune cell infiltration is pivotal in fibrosis, as these cells undergo cytoskeletal alterations to facilitate infiltration. Ca2+ is known to regulate cytoskeletal dynamics and migration in various cell types. Interestingly, we have a manuscript prepared for submission that demonstrates how dmXeB induces significant cytoskeletal alterations in invasive cancer cells, thereby reducing migration via an autophagy-mediated mechanism. Consequently, we anticipate that dmXeB may diminish immune cell infiltration, thus potentially contributing to a reduction in liver fibrosis. This has been further discussed in lines 402-412 of the revised manuscript.

Round 2

Reviewer 1 Report

Comments and Suggestions for Authors

None

Reviewer 2 Report

Comments and Suggestions for Authors

The authors responded positively to my concerns and have made the necessary revisions. The manuscript is now in good shape for acceptance.